# Review on the Regional Effects of Gastrointestinal Luminal Stimulation on Appetite and Energy Intake: (Pre)clinical Observations

**DOI:** 10.3390/nu13051601

**Published:** 2021-05-11

**Authors:** Jennifer Wilbrink, Gwen Masclee, Tim Klaassen, Mark van Avesaat, Daniel Keszthelyi, Adrian Masclee

**Affiliations:** 1Division of Gastroenterology-Hepatology, Maastricht University Medical Center, 6229 HX Maastricht, The Netherlands; jawilbrink@hotmail.com (J.W.); gwen.masclee@mumc.nl (G.M.); tim.klaassen@mumc.nl (T.K.); mark.van.avesaat@mumc.nl (M.v.A.); daniel.keszthelyi@maastrichtuniversity.nl (D.K.); 2NUTRIM School of Nutrition and Translational Research in Metabolism, 6229 ER Maastricht, The Netherlands

**Keywords:** intestinal brake, duodenal jejunal and ileal brake, tastants, energy intake, appetite, satiety, satiation, carbohydrate, protein, fat

## Abstract

Macronutrients in the gastrointestinal (GI) lumen are able to activate “intestinal brakes”, feedback mechanisms on proximal GI motility and secretion including appetite and energy intake. In this review, we provide a detailed overview of the current evidence with respect to four questions: (1) are regional differences (duodenum, jejunum, ileum) present in the intestinal luminal nutrient modulation of appetite and energy intake? (2) is this “intestinal brake” effect macronutrient specific? (3) is this “intestinal brake” effect maintained during repetitive activation? (4) can the “intestinal brake” effect be activated via non-caloric tastants? Recent evidence indicates that: (1) regional differences exist in the intestinal modulation of appetite and energy intake with a proximal to distal gradient for inhibition of energy intake: ileum and jejunum > duodenum at low but not at high caloric infusion rates. (2) the “intestinal brake” effect on appetite and energy appears not to be macronutrient specific. At equi-caloric amounts, the inhibition on energy intake and appetite is in the same range for fat, protein and carbohydrate. (3) data on repetitive ileal brake activation are scarce because of the need for prolonged intestinal intubation. During repetitive activation of the ileal brake for up to 4 days, no adaptation was observed but overall the inhibitory effect on energy intake was small. (4) the concept of influencing energy intake by intra-intestinal delivery of non-caloric tastants is intriguing. Among tastants, the bitter compounds appear to be more effective in influencing energy intake. Energy intake decreases modestly after post-oral delivery of bitter tastants or a combination of tastants (bitter, sweet and umami). Intestinal brake activation provides an interesting concept for preventive and therapeutic approaches in weight management strategies.

## 1. Introduction

After ingestion of food, the gastrointestinal (GI) tract is activated to facilitate transport, digestion and absorption of nutrients. Regional differences exist within the GI tract with respect to the modulation of these processes. Entry of nutrients into the small bowel activates so-called “intestinal brakes”, negative feedback mechanisms that not only affect motility and secretion but also appetite and energy intake.

Recent studies indicate that all macronutrients are able to activate these “intestinal brakes”, although to a different extent and through various mechanisms. In this review we provide a detailed overview of the current evidence with respect to four research questions: Are regional differences (duodenum, jejunum, ileum) present in the intestinal luminal modulation of appetite and energy intake?Is the “intestinal brake” effect on appetite and energy intake macronutrient specific?Is the “intestinal brake” effect that is observed in acute intervention studies maintained during repetitive activation?Can the “intestinal brake” effect on appetite and energy intake be activated via non-caloric tastants?

## 2. Nutrient Sensing in the Gut

Signals mediating satiety and satiation arise from various locations within the luminal gastrointestinal tract including the stomach, duodenum, jejunum, ileum and colon [1]. Ingestion of food results in mechanical stimulation by distension of the stomach and small intestine and in chemical stimulation via activation of nutrient receptors on enteroendocrine cells (EECs). These EECs play a pivotal role in the gastrointestinal and central regulation of not only of gastrointestinal (GI) motility and secretion but also of food intake. ECCs are scattered as single cells throughout the intestinal tract, located within the intestinal crypts and villi, and comprise about 1% of the total epithelial cell population. EECs act as sensors of luminal content, especially of nutrients, and function as trans-epithelial signal transduction conduits with apical physiochemical signals resulting in basolateral release and exocytosis of biological mediators. Nutrients or their breakdown products interact with G-protein coupled receptors (GPCRs) on EECs resulting in the secretion of gastrointestinal peptides such as cholecystokinin (CCK), peptide YY (PYY) and glucagon-like peptide-1 (GLP-1). These mediators either act in a classical endocrine fashion or by a paracrine effect on adjacent cells, including vagal afferent fibers. Non-nutrient chemical factors also regulate EEC activity, for example via sensing of tastants such as bitter, sweet, salt, sour and umami.

EECs carry specific receptors that upon sensing activate intracellular pathways either through direct gating of ion channels such as the sodium-dependent glucose co-transporter 1 (SGLT-1) or via activation of GPCRs. In recent years, various GPCRs have been identified such as sweet taste receptors (TAS1R/TAS2R), fatty acid-sensing receptors (GPR40, GPR43, GPR119 and GPR120) and various other types including PPAR, melanocortin, TRP family and opioids [2,3,4,5].

G-protein coupled taste receptors are expressed not only on the human tongue but also in stomach, proximal and distal small intestine and colon. Bitter taste is sensed by the TAS2R receptor while the TAS1R receptor family is triggered by sweet and umami. Taste receptors are able to “taste” luminal content and transmit signals that induce the release of GI peptides, thereby influencing satiety and food intake in humans [6].

## 3. Gastric Satiation Signals

Apart from its function to store food, to mix and grind stomach content and initiate the process of digestion, the stomach is able to monitor “food ingestion”. Consensus exists on the important role of gastric mechano-sensation in the regulation of satiety and food intake. In contrast to intestinal satiation, which is merely nutrient-induced, gastric satiation is merely volume-dependent. This has been shown for the first time in pyloric cuff experiments in rats [7]. In these experiments, saline or nutrient solutions were infused into the stomach but the pyloric cuffs prevented the infusate from entering the duodenum. Both the saline and nutrient infusate resulted in a similar reduction in food intake, showing that the nutritive effect did not add to the volumetric effect [7]. The satiating effect of gastric distension has been confirmed in human studies employing intragastric balloons. Prolonged distension of gastric balloons is known to result in reduction of food intake and subsequent weight loss [8].

## 4. Intestinal Satiation Signals

Exposure of the small intestinal lumen to nutrients induces satiety and a reduction in food intake. This was first observed in animal studies using gastric fistulas to exclude ingested food from entering the small intestine [9]. The animals would eat continuously when food was drained from the stomach. When the gastric fistula was closed and food entered the small intestine, the animals rapidly stopped eating, pointing to the pivotal role of the intestine in inducing satiety and satiation. Later, in vivo human intubation studies revealed that intestinal perfusion of proteins, carbohydrates or lipids, all resulted in a significant increase in satiety and decrease in food intake [10,11,12,13].

The GI peptides CCK, GLP-1 and PYY, secreted from EECs are known as mediators of intestinal satiation. These peptides induce their effects either via entering the bloodstream, acting as hormones (endocrine effect), via activation of vagal afferents (neuronal effect) or via an effect on neighboring cells (paracrine effect). Ghrelin is produced from the stomach and currently is the only GI peptide known to increase food intake by accelerating gastric emptying [1,6,13,14,15].

## 5. Intestinal Brakes

The process of motility, secretion, digestion and absorption is activated upon ingestion of food and its transport into stomach and duodenum. Thereafter, the appearance of nutrients further downward in the small intestine, during the process of digestion, results in activation of the so-called ”intestinal brakes”: feedback mechanisms from different parts of the intestine to the stomach, to the more proximal parts of the small intestine and also to the central nervous system. 

Entry of nutrients into duodenum or jejunum activates the “duodenal brake” or “jejunal brake” while infusion of nutrients into the ileum activates the more distal “ileal brake”.

The “duodenal brake” has been well documented as feedback from the duodenum to regulate gastric physiology, that is gastric acid secretion and gastric emptying. Inhibition of gastric emptying is a crucial “brake” against delivery of nutrients to the intestine in excess of digestive and absorptive capacity. In humans, gastric emptying is slowed in proportion to the energy density of the meal, thus leveling the rate of energy delivery to the duodenum [1,7,8,11].

The “ileal brake” is a negative feedback mechanism from the more distal to the proximal gastrointestinal (GI) tract that brings the process of transport, digestion and absorption of nutrients to an end (Figure 1). Activation of the “ileal brake” results in a reduction of gastric acid, biliary and pancreatic secretion, with inhibition of gastric emptying, intestinal motility and transport [14]. This concept of more distal intestinal brake activation with proximal inhibition of motility and secretion was derived from ileal transposition studies in rats. Koopmans and Sclafani were the first to show that transposing a segment of ileum to more proximal regions of the small intestine (i.e., duodenum) also resulted in a significant reduction in food intake and was associated with weight reduction in rats [15]. It was hypothesized that the hormonal changes induced by ileal transposition may have resulted in the observed reduction in food intake and increase in weight loss. Indeed, Strader et al. [16] observed that ileal transposition resulted in 3–4 times higher serum GLP-1 and PYY levels with an increase in satiety and weight loss that was proportional to the measured serum levels of GLP-1 and PYY.

Such a feedback inhibitory mechanism from the distal to the proximal GI tract has repeatedly been demonstrated in animal models. In the 1980s, the first human studies showed that infusion of fat or protein in the ileum delayed gastric emptying and intestinal transport and also increased feelings of satiety and reduced food intake. Welch et al., Read et al. and Spiller et al. [10,11,12,13] were among the first to point to the potent anorexic effect of “ileal brake” activation in humans through intestinal nutrient infusion. Welch et al. showed that an ileal lipid infusion of 370 kcal resulted in a decrease in food intake of 575 kcal, resulting in a net reduction in intake of 205 kcal. In a subsequent study comparing jejunal fat versus ileal fat infusion, the anorexic effect was more pronounced when fat was infused into the jejunum instead of the ileum [10]. One should consider that in these studies very high amounts of fat of up to 41 g were administered intestinally. These supraphysiologic amounts may have caused spilling of fat to more distal intestinal regions resulting in larger areas with activated nutrient receptors. 

## 6. Topics in Intestinal Brake, Appetite and Energy Intake

Several questions arise with respect to “intestinal brake” mechanisms and eating behavior:(1)Are regional differences (duodenum, jejunum, ileum) present in the intestinal luminal modulation of appetite and energy intake?(2)Is the “intestinal brake” effect on appetite and energy macronutrient specific? Are differences present between fat, carbohydrates and proteins?(3)Is the “intestinal brake” effect observed in acute intervention studies, maintained during repetitive activation?(4)Can the “intestinal brake” effect on appetite and energy intake be activated via non-caloric tastants?

### 6.1. Methods

Relevant studies for this review were identified by a PubMed search using search terms including duodenal, jejunal or ileal brake; duodenal, jejunal or ileal infusion; satiety, energy/food intake, or tastants. Only original articles involving human intervention studies and written in the English language were reviewed and selected. Additionally, reference lists of the original articles were reviewed for other relevant studies in order to be most complete in the current review. The included studies all relate to acute or short-term (single day to several days experiments) intervention studies in humans with either duodenal or jejunal or ileal intubation with intestinal perfusion of nutrients. In the reported publications, young, healthy volunteers, usually of male gender and aged 20–30 years with BMI in the normal range (20–25 kg/m^2^) have been studied. In the current review, we specifically report if studied participants were different (obese versus non-obese and younger versus older participants). Different types of stimuli have been used: long-chain fats and fatty acids, protein or amino acids and carbohydrates. The type of stimuli that have been used are listed in Table 1, Table 2 and Table 3 and for each study separately in the Appendix A: Table A1, Table A2 and Table A3.

### 6.2. Topic 1: Site Specific Effects on Food Intake and Satiety: Duodenum-Jejunum-Ileum

The human studies evaluating the intestinal brake effects have used intubation techniques to isolate the intestinal effects from oral or intragastric effects. Most studies with intestinal nutrient administration have focused on duodenal delivery. Duodenal positioning of an intestinal tube is more easily performed and more convenient compared to intubation of jejunum or ileum. Satiety and reduction in energy intake of a meal during and after intestinal nutrient infusion are the two relevant outcome parameters. The energy content of the nutrient infusate has also been taken into account. The net effect on energy intake has been calculated as the reduction in energy intake of the meal minus the energy intake via the infusate. In case of more distal delivery of nutrients, digestion and absorption may not be complete so that the net reduction in energy intake may have been underestimated in the conditions of the currently included studies.

#### 6.2.1. Dietary Fat: Site Specific Effects?

Compared to oral fat intake, ileal infusion of the same amount of fat (6 g) has a significantly more pronounced effect on food intake resulting in a 15% reduction in caloric intake of a subsequent meal [27]. Maljaars et al. repeated this experiment with 3 g fat and confirmed the observation of an ileal feedback of fat on eating behavior to be operative even at very low doses of fat [28]. It is not only the amount of fat but also the physicochemical properties of fat that affect the magnitude of the inhibitory effect on food intake and satiety. The effect on satiety parameters appears to be more pronounced with smaller fat droplet sizes, in the duodenum but also in the ileum. The reduction in food intake was 9% higher after fine versus coarse droplet infusion, both for duodenal and ileal fat delivery [32]. The reduction in hunger scores and food intake was more pronounced with increasing fatty acid chain length of intraduodenally administered fatty acids [46,47]. Maljaars et al. showed that intra-ileal triacylglycerols with unsaturated fatty acids resulted in a more pronounced increase in satiety compared to triacylglycerols with saturated fatty acids [47].

Data of studies on intestinal site-specific effects of dietary fat on eating behavior are shown in Table 1. After *duodenal* administration of fat at low infusion rates of 0.25 to 1.5 kcal per min, the inhibitory effect on energy intake is very small (mean 3%, range 0–8%) without a significant effect on fullness or satiety. At higher fat infusion rates of 2–4.9 kcal per min, a reduction in energy intake of 21% (range 10–32%) with a significant increase in satiety parameters was observed.

Infusion of fat into the *jejunum* at a high dose of fat of almost 5 kcal per min leads to a mean reduction in energy intake of 31% (range 12–50%) and a significant increase in satiety that at an equicaloric load is more pronounced in the *jejunum* compared to the *duodenum*.

For *ileal* fat administration, low fat infusion rates of around 0.5 kcal per min result in significant reductions in subsequent energy intake of 18% (range 15–21%) and increases in satiety. At higher *ileal* fat infusion rates of up to 5 kcal per min the effect on energy intake is even more pronounced with 31% reduction (range 30–32%).

These studies reveal that at low infusion rates the inhibitory effect of fat on energy intake is more pronounced in the ileum compared to jejunum and duodenum. When the infusion rate of fat is high, at doses of 5 kcal per min, the inhibitory effect on energy intake is in the same range for duodenal, jejunal and ileal fat delivery.

With respect to appetite: in the study of Maljaars et al. [28] the lower and higher dose of ileal fat of 3 and 9 g respectively, resulted in a similar reduction in appetite and increase in satiety, without any evidence for dose dependency, in contrast to intraduodenal fat. This difference between duodenum and ileum may be related to lipolytic capacity that is much smaller in the ileum compared to the duodenum. Digestion of triacylglycerol to fatty acids is considered a necessary step for fat to induce its satiety-inducing effects [48].

Concerning infusion of fatty acids, only data for duodenal perfusion are available. At equicaloric infusion rates, the inhibitory effect on food intake is more pronounced with fatty acids compared to fat: infusion ranges 0.4–0.75 and 0.3–0.9 kcal per min respectively result in reductions in food intake of 10–15% with fatty acids and 0% with fat.

With respect to reduction of energy intake, it is essential to take into account the energy content of the nutrients perfused. In the Table A1, we provide individual study data on the caloric content of the fat and fatty acid nutrients infused and of the change in energy intake (kcal) of the meal compared to control condition, per study [10,13,19,20,22,24,26,27,29,30,31,32,33,34,35,36,37]. The net effect is the reduction in energy intake of the meal minus the energy intake via the infusate.

#### 6.2.2. Dietary Proteins: Site Specific Effects?

In general, proteins are known to be more satiating at an equicaloric basis compared to either lipids or carbohydrates. Proteins are therefore considered to be the most anorectic of the three macronutrients. Data of studies on intestinal site-specific effects of dietary proteins on eating behavior are shown in Table 2. Several groups have evaluated the effects of *intraduodenal* protein administration on food intake and satiety. Ryan et al. and Soenen et al. [24,35,36] have performed *intraduodenal* perfusion studies with whey protein. A dose-response effect was observed with respect to reductions of energy intake. Infusion doses of 0.5, 1.5 kcal and 3 kcal per min resulted in stepwise dose-dependent inhibitions in subsequent energy intake of 6%, 12.5% and 21% respectively. *Jejunal* and *ileal* protein infusion resulted in reductions in energy intake of respectively 9% and 22% at a dose of 0.85 kcal/min. In the *ileum* lower infusion rates of 0.19–0.57 kcal per min already resulted in a reduction in energy intake of 9.9–14%. Thus, also for proteins an inhibitory, dose-dependent, effect of intestinal protein infusion on energy intake has been demonstrated with a proximal to distal gradient.

Remarkably, no significant effect on fullness, hunger or satiety during or after *intraduodenal* protein infusion was observed (Table 2). Only during *ileal* protein infusion, a significant reduction of hunger was noted. An explanation for the lack of effect of intestinal proteins on satiety parameters may be related to differences in the appetite suppressing effects within different sources of proteins [49].

Concerning amino acids: at low *duodenal* infusion rates of 0.07–0.15 kcal per min and of 0.2–0.4 kcal per min, the reductions in energy intake were small, respectively 5% and 13%. All studies included a 90 min duration of infusion. At equicaloric infusion rates, the inhibitory effect on energy intake of proteins and amino acids was in the same range. Amino acids studied included L-tryptophan [31,38] and leucine [39]. Across studies, the inhibitory effect on energy intake between the different amino acids showed similar effect sizes (net reduction 175 ± 89 kcal [31]; 206 ± 68 kcal [38]; 170 ± 48 kcal [39], however direct comparison of the effect on energy intake reduction between different amino acids has not been studied yet.

In the Table A2 we provide individual study data on the caloric content of the protein and amino acid nutrients infused and of the change in energy intake (kcal) of the meal compared to control condition, per study [29,32,35,39,40,41,42,43]. The net effect is the reduction in energy intake minus the energy intake via the infusate.

#### 6.2.3. Dietary Carbohydrates: Site Specific Effects?

The existence of a “duodenal brake” for carbohydrates with satiating effects has been well established. Lavin et al. [43] observed a significant reduction in energy intake and suppression of hunger when glucose was infused into the *duodenum* compared to the same amount of glucose administered intravenously.

Data of studies on intestinal site-specific effects of dietary carbohydrates on eating behavior are shown in Table 3. Infusion of glucose into the *duodenum* at doses of 0.66 to 2 kcal per min reduced energy intake non-significantly by 10% (range 5–13%). At higher *duodenal* infusion rates a significant inhibition of energy intake was found starting from a caloric load of 2.86 kcal glucose per min resulting in a mean 17% reduction (range 11–26%) in energy intake.

Starting from an *intraduodenal* caloric glucose load of 2.0 kcal per min the “desire to eat” and hunger were suppressed. As shown in Table 3, a dose-dependent effect of intestinal glucose on both satiety and energy intake has been observed. Changes in blood glucose concentrations have significant impact on gastric emptying. A delay in gastric emptying may increase satiation and reduce energy intake. Thus, blood glucose levels should be monitored and adjusted in patients with pronounced hyperglycemia as in diabetes mellitus.

Infusion of glucose at a 1 kcal per min rate into the *jejunum* did not reduce but increased energy intake compared to infusion of the same amount of glucose to the *duodenum*. In the *intrajejunal* experiment, food intake was 11% higher compared to the *duodenal* experiment [42]. Possibly in the duodenal experiment, a larger intestinal area has been exposed to glucose and may have resulted in a more pronounced reduction in energy intake.

Infusion of glucose in the *ileum* at a dose of 0.66 kcal per min induced a reduction of energy intake of 10%, a result that is in line with the reduction of energy intake when the same amount of glucose is administered into the *duodenum*. When instead of glucose, sucrose is administered in low doses of 0.19 to 0.57 kcal per min the inhibitory effect on energy intake is even more pronounced: 21% and 32% respectively. Satiation and fullness were not affected, neither by sucrose nor by glucose.

In the Table A3 we provide individual study data on the caloric content of the carbohydrate nutrients infused and of the change in energy intake (kcal) of the meal compared to control condition, per study [23,26,27,28,32,45,46,48,49]. The net effect is the reduction in energy intake minus the energy intake via the infusate.

### 6.3. Topic 2: Is the Intestinal Brake Effect on Appetite and Energy Intake Macronutrient Specific?

This question was addressed in a study by van Avesaat et al. [29] directly comparing isocaloric infusion of fat, protein and carbohydrate with safflower oil, casein and sucrose respectively into the ileum. A significant reduction in energy intake was observed with all three macronutrients: for sucrose 32%, for fat 21% and for casein 22% (differences between macronutrients: n.s.) These results indicate that equicaloric amounts of macronutrients induce an ileal brake inhibition of energy intake and affect eating behavior to the same extent. 

In Table 4 we present summarized data from all the studies that are presented in Table 1, Table 2 and Table 3. We specify in Table 4 energy intake in response to fat, carbohydrates and protein but now compared for each location of perfusion: duodenum, jejunum and ileum. Note that the reduction in energy intake is presented as percentage reduction of energy intake as compared to control conditions and does not represent the absolute difference in caloric intake, as the caloric load by infusion of the macronutrient has not been taken into account in this calculation.

For *duodenal* perfusion, the responses to the three macronutrients, when based on caloric perfusion rate, were of the same magnitude: at caloric loads of 0–1 kcal per min, the energy intake was reduced by max 10% while at caloric loads of around 3 kcal per min the energy intake was reduced more than 20%.

For *jejunal* perfusion: only few data are available. The magnitude of energy intake reduction of a subsequent meal was more pronounced after high kcal infusion of fat compared to low kcal dose of protein while in the glucose infusion experiment at 1 kcal per min jejunal infusion resulted in a higher energy intake compared duodenal infusion of the same glucose load.

With respect to the *ileum*, it appears that even at low infusion rates of up to 1 kcal per min the reduction in energy intake is more pronounced with 10–30% inhibition compared to a maximum of 10% for *duodenal* brake and max 9% for *jejunal* brake. At doses > 3 kcal per min the magnitude of energy intake reduction is comparable for duodenal, jejunal and ileal brake with 20–30% reductions in energy intake. Taken together, these data point to brake effects that appear to be not so much macronutrient specific but more dependent on caloric loads.

With respect to location, the data point towards more pronounced brake effects for the ileum compared to jejunum and duodenum (Table 4): a distal to proximal gradient that is equal for the three macronutrients. We separately analyzed the studies [10,19,32,37,42,44] that directly compared the effect of region of infusion of a macronutrient on energy intake within the same study (Table 5) because this represents the most valid comparison. The results of these separately analyzed six studies are not different from the results of the combined data of all published studies together listed in Table 1, Table 2, Table 3 and Table 4.

Additionally, an analysis has been performed considering the net energy intake reduction, that is, the reduction in energy intake (kcal) of the meal minus the caloric content of the nutrient infusion. This means that energy intake reduction resembles the absolute reduction in intake of energy from a meal, thus the amount of meal not eaten, while taking the amount of infusion into account. For this analysis, the data from all the individual studies presented in the Table A1, Table A2 and Table A3 have been used. These data are presented in Table 6 at an aggregated level. In the studies with duodenal delivery of nutrients high caloric nutrient loads have been used, much higher compared to jejunal or ileal delivery of nutrients. While a reduction in energy intake was observed after duodenal infusion of carbohydrates, fat and proteins as shown in Table 4, when taking into account the high caloric load of the perfused duodenal nutrients, no net reduction was observed but an increase in the amount of calories ingested (Table 6) for duodenal delivery of fat and carbohydrates. For duodenal protein infusion, the net reduction in energy intake remains substantial. When comparing net intake reduction after ileal infusion, the reduction is in the same range for fat, protein and carbohydrate based on caloric load. Future studies on intestinal brake mechanisms should take into account more systematically the caloric load delivered with intestinal nutrient infusion. In case of more distal delivery, digestion and absorption may not be complete so that the net reduction in energy intake may have been underestimated in the conditions of the currently included studies.

Lin et al. and Meyer et al. [50,51] have shown that increasing the small intestinal area exposed to nutrients resulted in more potent brake effects on gastric emptying and on satiety. Maljaars et al. [27] have investigated in more detail whether exposure of larger intestinal areas to nutrients causes a more potent effect on satiety and food intake. In three different experiments the same amount of fat (6 g in total) was administered at equal perfusion rates of 0.6 kcal per min for 90 min into (a) ileum only (6g) (b) duodenum (2g) jejunum (2g) and ileum (2g) simultaneously or (c) duodenum (2g) jejunum (2g) and ileum (2g) sequentially. Compared to control condition with oral fat, inhibition of food intake was 8% and 4% resp. for the simultaneous and sequential perfusion of larger intestinal areas while perfusion of the ileum resulted in the most pronounced and statistically significant reduction in food intake of 16%. Thus, increasing the small intestinal area did not result in larger reduction of food intake. Compared to control condition, hunger was significantly reduced during all three experiments. 

### 6.4. Topic 3: Is the Acute “Intestinal Brake” Effect Maintained during Repetitive Activation?

Most of the studies performed so far have evaluated acute intestinal brake interventions on energy intake and satiety but did not explore whether the observed reductions in food intake in the acute experiments persist after repetitive activation. Data on chronic, prolonged jejunal-ileal brake activation have been obtained with bariatric surgery especially with malabsorptive procedures such as Roux-en-Y Gastric Bypass (RYGB). In this combined restrictive and malabsorptive bariatric procedure the proximal small intestine is bypassed and food is delivered to the more distal small intestine resulting in significant food intake reduction and weight loss on the long term [52].

Avesaat et al. [53] were among the first to investigate the effect of repeated, four days, activation of the “ileal brake” with ileal protein infusion on energy intake and satiety but also on gut peptide secretion and gastric emptying. Compared to control condition, energy intake during brake activation with proteins was lower: respectively 7%, 9%, 17% and 10% at days 1, 2, 3, and 4 compared to control (differences versus control condition: n.s.). While food intake was not significantly affected, satiety parameters were significantly increased and gastric emptying was delayed. These effects did not change during the four days of repetitive ileal brake activation.

### 6.5. Topic 1–3: Intestinal Brake to Nutrients: Summary of Findings and Perspectives

Compared to the duodenal brake, activation of the jejunal and ileal brake results in a somewhat more pronounced effect on energy intake and satiety, pointing to a distal to proximal gradient in intestinal brake efficacy. This distal to proximal gradient effect remained after correction for the caloric load of the nutrients infused. Thus, the net effect of ileal and jejunal brake activation on energy intake reduction is larger compared to duodenal brake effects (Table 6). During repetitive activation of the ileal brake for a maximum of four days, no adaptation or reduction in brake efficacy was observed.

Several limitations of the human studies with brake activation should be mentioned. First, all intestinal brake activation studies in humans have been performed with intestinal intubation that causes discomfort and inconvenience and may negatively affect eating behavior. On the other hand, the intervention and control experiments all have been performed during intestinal intubation so that conditions were equal. Second, one should realize that the magnitude of the brake effect on food intake is rather small. At lower infusion rates an intake reduction of a subsequent meal of about 10% is reached. Although this effect is not significantly different from control condition it is very consistently reported in all studies and it is clinically relevant as this may help in subsequent desired weight loss. Third, future studies should take into account more systematically the caloric load of the nutrients perfused.

The question arises whether the “ileal brake” can be activated without the need for intestinal intubation. In this respect encapsulation of nutrients or use of slow-release formulas are interesting alternatives. In order to reach the ileum before being digested and absorbed, nutrients should be protected by a structure that not only survives the acidic conditions of the stomach but also protects against the action of digestive enzymes and bile in the proximal small intestine. A prerequisite is that lipolysis and proteolysis should not be completely inhibited in the small intestine, but are delayed to prevent early absorption as degradation products of lipid, protein or carbohydrate digestion. When considering whether one specific macronutrient would be most suitable for encapsulation one should keep in mind the effect on energy intake reduction from the different nutrients (Table 4 and Table 6). The effect of reduction of energy intake is dependent on the location of release (ileal versus duodenal) rather than on a specific nutrient, as for ileal release all nutrients appear to have similar effects.

With Fabuless, an emulsion of fractionated palm and oat oils dispersed in water thought to active the jejunal-ileal brake, initially a reduction in energy intake was observed [54], but subsequent studies failed to substantiate this effect [55]. Corstens et al. [56,57] have applied food graded micro-encapsulation systems to study ileal brake activation and satiety induction via delayed lipolysis. A human intervention study was performed using encapsulated lipid as emulsion-alginate beads or an equicaloric mixture of the same non-encapsulated nutrients with similar sensory properties as control condition [56]. Food intake of a subsequent meal was significantly reduced (intake 770 ± 38 kcal versus 821 ± 40 kcal; *p* = 0.016) and satiety was significantly increased after intake of the active substance [56]. Again, the reduction in caloric intake was only small, 6–7%, but may be clinically relevant. Alleleyn et al. [58] have studied the effect of an encapsulated carbohydrate-protein mixture on meal intake in healthy volunteers. A small but significant reduction of 6% in caloric meal intake was observed. These first human intervention studies need confirmation and the application of beads to modulate food intake should be evaluated in larger scale and longer-term intervention studies. With respect to encapsulated carbohydrates in the form of sugar: more distal (ileal) delivery of sugars (glucose) will lead to an increase in GLP-1 release with subsequent insulin release and increase in insulin sensitivity, factors that are beneficial in overweight and type 2 diabetes mellitus. In general, for encapsulation, the more “energy-dense” compounds (fat, fatty acids) are of particular interest.

Several questions arise: Can this reduction in caloric intake be repeated over the day during every meal? Is it affecting in-between meal snacking? Is the effect maintained over a longer period of time? What is the overall effect on weight regulation? These questions, including safety issues, need to be addressed in future studies. 

### 6.6. Topic 4: Can the “Intestinal Brake” Effect on Appetite and Energy Intake Be Activated via Non-Caloric Tastants?

The five prototypical basic tastes sweet, salt, sour, bitter and umami are sensed by taste buds present on the tongue. Ion channels mediate the sensing of salty and sour taste while sensing sweet, bitter and umami taste is mediated by two families of taste receptors. Taste receptor family 1 (TAS1R) generally senses sweet and umami taste and taste receptor family 2 (TAS2R) primarily senses bitter taste [6]. It has been stated that these prototypical tastes exist in order to predict the type of food that will be ingested (i.e., sweet for saccharides, umami for glutamate, and bitter for potentially toxic substances). However, it is well known that negative affective responses to bitter can be uncoupled and converted into positive responses, as for caffeine [59,60]. Taste receptors are not only present on the tongue but are expressed throughout the entire human gut [61,62], in particular on EECs. Activation of taste receptors can result in the release of GI peptides such as CCK, PYY and GLP-1, known to influence satiety and eating behavior. Thus, activation of taste receptors can be elicited using non-caloric tastants. This concept of non-caloric modulation of satiety and eating behavior via intestinal taste receptors deserves further evaluation. Recently several smaller-scale clinical studies have been published on this topic focusing on gastrointestinal delivery of tastants. In these studies, the hypothesis was tested that post-oral delivery of non-caloric tastants will result in a net decreased energy intake compared with placebo. Klaassen et al. recently published a systematic review and meta-analysis on this topic [63]. These authors report on the effects of gastrointestinal administration of tastants on eating behavior. For sweet taste, aspartame and rebaudioside A have been used (Table 7) and a reduction in energy intake varying from 0–10% was observed.

The effect of gastrointestinal delivery of bitter tastants has been studied more extensively. Seven studies showed, small to moderate, non-significant reductions in food intake of a subsequent meal varying from 5 to 11%. In one study with repetitive intraduodenal administration of a bitter mixture, overall daily food intake was significantly reduced by 22% (340 kcal).

Umami after intraduodenal administration of monosodium glutamate, did not affect intake of the subsequent meal compared to placebo. When sweet, bitter and umami tastants were infused simultaneously, subsequent caloric food intake was significantly impaired by 14%, compared to control condition [66]. None of these tastants when administered separately, had a significant effect on caloric intake. In a subsequent study, with duodenal and ileal infusion of the same combination of sweet, bitter and umami, these findings could not be confirmed [74].

The currently available data show that, among tastants, bitter compounds appear to be the most effective in influencing eating behavior. Energy intake, in the acute setting, decreased modestly after post-oral delivery of bitter tastants. Future studies should focus on dosing of tastants and on potential mechanisms of action. In this respect, effects on GI motility and on systemic and local GI peptide secretion have been observed [70,72,75,76]. Current knowledge on the effects of tastants on energy intake and satiety is limited. The most appropriate location(s) for tastant delivery to modulate eating behavior remains to be established. More research investigating the delivery of various tastants to different locations in the GI tract is needed.

## 7. Conclusions

In this study, we have reviewed the current literature with respect to human intervention studies of intestinal feedback mechanisms on appetite and energy intake. Most human studies have been performed with intestinal intubations and infusion of nutrients. Recent evidence indicates that:

(1)Regional differences exist in the intestinal modulation of appetite and energy intake with a distal to proximal gradient for inhibition of energy intake: ileum and jejunum > duodenum. This distal to proximal gradient effect remains after correction for the caloric load of the nutrients infused.(2)The “intestinal brake” effect on appetite and energy appears not to be macronutrient specific. At equicaloric amounts, the inhibition on energy intake and appetite is in the same range for fat, protein and carbohydrate.(3)Data on repetitive ileal brake activation are scarce because of the need for prolonged intestinal intubation. During repetitive activation of the ileal brake for up to 4 days, no adaptation was observed but overall, the inhibitory effect on energy intake was small.(4)The concept of influencing energy intake by intra-intestinal delivery of non-caloric tastants is intriguing. Thus far, the available data show that, among tastants, bitter compounds appear to be more effective in influencing energy intake. Energy intake, in the acute setting, decreased modestly after post-oral delivery of bitter tastants or a combination of tastants (bitter, sweet and umami). An advantage is that tastants are non-caloric, in contrast to nutrients. Future studies should focus on optimal dosing and delivery of tastants and their mechanisms of action.

Intestinal brake activation provides an interesting concept for preventive and therapeutic approaches in future weight management strategies.

## Figures and Tables

**Figure 1 nutrients-13-01601-f001:**
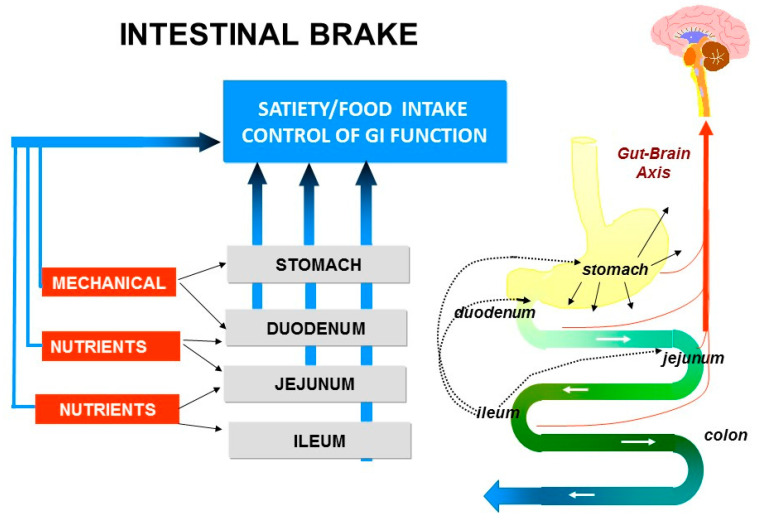
Intestinal brake: effect of luminal stimulation in stomach, duodenum, jejunum and ileum on neurohormonal control of gastrointestinal function, food intake and satiety.

**Table 1 nutrients-13-01601-t001:** Effect of infusion of fat or fatty acids into duodenum, jejunum or ileum on energy intake and satiety.

Location	Infusate and Infusion Rate: Kcal per min	Energy Intake		Satiety ↑ ↓	References
		% Reduction	↑ ↓		
Fat					
Duodenum	1.1 (0.25–1.5) Corn oil, safflower oil, Intralipid	3% (0–8)	=	=-↑	[17,18,19]
	3.3 (2.0–4.9) corn oil, Intralipid	21% (10–32)	↓↓	↑	[10,17,18,20,21,22,23,24,25]
Jejunum	4.9 corn oil	12%	↓↓	↑↑	[26]
	4.9 corn oil	50%	↓↓↓	↑↑	[10]
Ileum	0.5–0.6 rapeseed, safflower oil	18% (15–21)	↓↓	↑	[19,27,28,29]
	1.8–4.9, corn oil, safflower oil	31% (30–32)	↓↓↓	↑↑	[10,13,28]
**Fatty acids**					
Duodenum	0.2–0.3 lauric acid	0–4%	=-↓	=	[30,31]
	0.4–0.75 lauric, LCFA	10–15% (60%) *	↓↓	↑	[30,32,33,34]
Jejunum	-	-	-	-	-
Ileum	-	-	-	-	-

Energy intake in % reduction in caloric intake: = no reduction or increase ↓ 0–10% reduction, ↓↓ 10–25% reduction, ↓↓↓ > 25%. reduction versus control condition; ↑ 0–10% increase, ↑↑ 10–25% increase * = in one study (19) a reduction in energy intake of 60% was observed, but subjects were severely nauseated. Data on net intake are not corrected for caloric content of the nutrient infusate. LCFA: long-chain fatty acids. Energy intake reduction is presented as percentage reduction of energy intake as compared to control condition.

**Table 2 nutrients-13-01601-t002:** Effect of infusion of protein or amino acids into duodenum, jejunum or ileum on energy intake and satiety.

Location	Infusate and Infusion Rate Kcal per min	Energy Intake		Satiety	References
		% Reduction	↑ ↓		
**Protein**					
Duodenum	0.5–1.5 whey, casein	8%	↓	=	[35,36,37]
	3.0 whey, casein	21%	↓↓	=	[24,35,36]
Jejunum	0.85 casein	9%	↓	=	[37]
Ileum	0.19 casein	9.9%	↓	=	[29]
	0.57–0.85 casein	14–22%	↓↓	↑↑	[29,37]
**Amino acids**					
Duodenum	0.07–0.15 tryptophan	5%	↓	↑	[31,38]
	0.2–0.4 tryptophan, leucine	13%	↓↓	=	[38,39]
Jejunum	-	-	-	-	-
Ileum	-	-	-	-	-

Energy intake in % reduction in caloric intake: = no reduction or increase, ↓ 0–10% reduction, ↓↓ 10–25% reduction, ↓↓↓ > 25% reduction. Satiety: = no reduction or increase, ↑ 0–10% increase, ↑↑ 10–25% increase. Energy intake reduction is presented as percentage reduction of energy intake as compared to control condition.

**Table 3 nutrients-13-01601-t003:** Effect of infusion of carbohydrate (glucose or sucrose) into duodenum, jejunum or ileum on energy intake and satiety.

Location	Infusate and Infusion Rate Kcal per min	Energy Intake		Satiety	References
		% Reduction	↑ ↓		
Duodenum	0.6–2.0 glucose	10% (5–13)	↓	↑	[40,41,42]
	2.9–4.0 glucose	17% (11–26)	↓↓	↑↑	[21,22,23,40,41,42,43,44,45]
Jejunum	1.0 glucose	+11% *	↑↑	=	[42]
Ileum	0.19 sucrose	21%	↓↓	=	[29]
	0.57 sucrose	32%	↓↓↓	=	[29]
	0.66 glucose	10%	↓	=	[44]

Energy intake in % reduction in caloric intake: = no reduction or increase, ↓ 0–10% reduction, ↓↓ 10–25% reduction, ↓↓↓ > 25% reduction; + 11% * = increase in intake jejunal compared to duodenal infusion [42]. Energy intake reduction is presented as percentage reduction of energy intake as compared to control condition. Satiety: = no reduction or increase, ↑ 0–10% increase, ↑↑ 10–25% increase in satiety

**Table 4 nutrients-13-01601-t004:** Comparison of Energy Intake reduction (EI-red) in response to infusion of equicaloric amounts of fat, protein or carbohydrate (infusion rate in kcal/min) per location: duodenum, jejunum or ileum. Combined results of data obtained from published studies (see references).

Location	Fat		Carbohydrate		Protein	
	kcal/min	EI-Red	kcal/min	EI-Red	kcal/min	EI-Red
Duodenum	0.25–1.5	0–15%	0.6–2	5–13%	0.5–1.5	6–13%
	2–5	10–32%	2.86–4	11–26%	3.0	21%
Jejunum	4.9	12–50%	1	+11%	0.85	9%
Ileum	0.5–0.6	15–21%	0.19–0.66	10–32%	0.19–0.85	14–22%
	1.8–4.9	30–32%				

For duodenum, jejunum, ileum: equicaloric intake reduction: fat = carbohydrate = protein Duode-num/jejunum: infusion rate < 1 kcal/min: intake reduction < 10%, Duodenum/jejunum: infusion rate > 3 kcal/min: intake reduction > 20%, Ileum: infusion rate < 1 kcal/min: intake reduction 10–32%, ileum: infusion rate > 3 kcal/min: intake reduction > 30%. Reference: Duodenum: Fat: [10,17,18,19,20,21,22,23,24,25,30,31,32,33,34]. Protein: [23,31,35,36,37,38,39]. Carbohydrate: [21,22,23,40,41,42,43,44,45]. Jejunum: Fat [10,26]. Protein [37]. Carbohydrate: [42]. Ileum: Fat: [10,13,19,27,28,29]. Protein: [29,37]. Carbohydrate: [29,44]. Note that + under EI-red means an increase in energy intake. Energy intake reduction is presented as percentage reduction of energy intake as compared to control condition.

**Table 5 nutrients-13-01601-t005:** Comparison of net effect of energy intake reduction: reduction in energy intake of a meal minus energy content of infusate of fat, proteins or carbohydrates with comparison per location within the same study.

Reference	Location	Infusate		Reduction in Energy Intake (EI) of Meal	Net Effect: Reduction EI Meal-EI Infusate
		Type	Energy Content of Infusate		
41	Duodenum	casein	60 kcal	+20 kcal	−
	Jejunum	casein	60 kcal	40 kcal	−
	Ileum	casein	60 kcal	80 kcal	+
47	Duodenum	glucose	90 kcal	+160 kcal	−
	Jejunum	glucose	90 kcal		−
48	Duodenum	glucose	56 kcal	58 kcal	=
	Ileum	glucose	56 kcal	119 kcal	+
10	Jejunum	corn oil	370 kcal	1100 kcal	++
	Ileum	corn oil	370 kcal	650 kcal	++
33	Duodenum	rapeseed oil	54 kcal	14 kcal	−
	Ileum	rapeseed oil	54 kcal	18 kcal	−
19	Duodenum	canola oil	54 kcal	ileum vs. duo:	
	Ileum	canola oil	54 kcal	76 kcal	+

− means reduction EI meal < EI infusate. = means reduction EI meal = EI infusate. +/++/+++ means reduction EI meal > EI infusate. +: 0–100 kcal; ++100–300 kcal; +++: > 300 kcal.

**Table 6 nutrients-13-01601-t006:** Comparison of net energy intake reduction in response to infusion of fat, fatty acids, proteins, amino acids or carbohydrate per location: duodenum, jejunum or ileum.

Location	Infusate	Infusate	Infusate	Infusate	Infusate
	LCT fat	LC fatty acids	Proteins	Amino acids	Carbohydrates
Duodenum	−	+	+	+	−
Jejunum	++	NA	+	NA	−
Ileum	++	NA	++	NA	++

Combined results of data obtained from published studies listed in Table 1, Table 2 and Table 3 and individual data from Table A1, Table A2 and Table A3. NA: not assessed. Net reduction: reduction in energy intake of meal minus energy content of infusate. − means net reduction in energy intake is negative: increase in the amount of calories ingested. = means no net reduction in energy intake, reduction in energy intake of meal = energy content of infusate. +/++/+++ means net reduction in energy intake: reduction in energy intake of meal > energy content of infusate + net reduction 0–100 kcal ++ net reduction 100–300 kcal +++ net reduction > 300 kcal.

**Table 7 nutrients-13-01601-t007:** Effect of tastants after duodenal, jejunal or ileal delivery on energy intake.

Taste	Tastant	Administration	Energy Intake Reduction		Reference
			kcal	%	
Sweet	aspartame	gastric capsule		10%	[64]
	aspartame	gastric capsule		0%	[65]
	rebaudioside A	duodenal tube	26 kcal	5%	[66]
Bitter	quinine	acid resistant capsule			[67]
	quinine	duodenal tube	44 kcal	9%	[66]
	secoiridoids	micro encapsulation	88 kcal	11%	[68]
			340 kcal (day)	22%	
	bitter mixture	gastric capsule	109 kcal	7%	[69]
	denatonium bezoate	gastric tube	76 kcal	9.5%	[70]
	quinine 600 mg	gastric tube	53 kcal	5%	[71]
	quinine 275 mg	gastric tube	+26 kcal	+3%	[71]
	quinine	gastric tube	68 kcal	9%	[72]
	quinine 37.5 mg	duodenal tube	31 kcal	3%	[73]
	quinine 75 mg	duodenal tube	59 kcal	5%	[73]
	quinine 225 mg	duodenal tube	11 kcal	1%	[73]
Umami	monosodium glutamate	duodenal tube	+5 kcal	+1%	[66]
Combination: sweet, bitter and umami	Reb A, quinine and MSG	duodenal tube	64 kcal	14%	[66]
	Reb A, quinine and MSG	duodenal tube	17 kcal	+2%	[72]
		ileal tube	28 kcal	+4%	[74]
		duodenal+ ileal tube	31 kcal	+4%	[74]

RebA: rebaudioside A, MSG: monosodium glutamate. Note that + means an increase in energy intake. Energy intake reduction is presented as percentage reduction of energy intake as compared to control conditions.

## Data Availability

Data for this review have been obtained from published studies regarding the topics, not from databases.

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
