# Peer review of "Review on the Regional Effects of Gastrointestinal Luminal Stimulation on Appetite and Energy Intake: (Pre)clinical Observations"

_nutrients, 2021, doi:10.3390/nu13051601_

Round 1

Reviewer 1 Report

This review aims to summarize the literature on regional effects of GI stimulation on the negative feedback mechanisms that control gastric emptying and gut motility, and appetite and energy intake. The authors look at effects of region, potential differences of nutrients on EI suppression, and the use of non-caloric tastants to suppress energy intake. The review is concise, addresses clearly stated research questions, and is well written. The review contributes novel data to the literature on the gastrointestinal regulation of appetite and energy intake.

I have a few comments that the authors may want to consider.

  1. One major issue is that, although the authors state that a suppression of energy intake as low as 6-7% is clinically relevant, they do not consider the energetic content of the nutrient infusions itself in their statements. If the energy content of the nutrient infusion is identical to, or larger than the suppression in energy intake at the subsequent meal, the strategy to ‘preload’ people with nutrients aiming to reduce their energy intake is arguably not feasible as a potential weight management strategy. Could the authors please include statements on the difference in suppression between the energy content of the infusion and the suppression of energy intake in the studies they included?
  2. The authors compare studies with very different protocols to describe their findings. Would the authors consider to include a statement on the nature of the studies included? For example, did the papers include similar participant groups in terms of health, gender, age – and if not, were there differences in findings between the outcomes of the studies? For example, older and obese participants may be less sensitive to the suppressive effect of nutrient infusion on EI, and therefore, may show different results from studies performed in younger/healthy participants. Furthermore, how homogenous are the studies in terms of interventions used? For example, did the described studies use the same sources of proteins/fat for infusion or not (the authors mention this for carbohydrates but not protein, AA, and fat)?
  3. Furthermore, there are some studies that directly compare the effect of region of infusion on energy intake (ref 10, 41, 47, 48 – maybe others?) within their study. Do these studies show differences between regions of infusion on suppression of EI? Would the authors consider to briefly mention these studies in the text, as these studies would have used the same study protocols for each infusion region, and therefore, may be more appropriate to compare effects between regions than the comparison of studies with different protocols, nutrients loads and/or interventions?
  4. Figure 1: this figure is not entirely clear to me. Where does the arrow to the colon refer to? And should the arrow from the ileum point to the word ‘duodenum’?
  5. Lines 237-240: Which amino acids were looked at in the papers included? Are there any differences between the AA’s included?
  6. Lines 362-369: the authors talk about the opportunities for applications of their findings. From their findings, are there any nutrients that are more relevant to encapsulate than others? For example, the delivery of carbohydrates, especially in the form of sugar, may not be suitable as it may come with potential negative effects on blood glucose regulation, especially in those who are at risk to develop diabetes. On the other hand, non- or low- caloric compounds may be more feasible as the difference between the caloric load of the compound delivery and caloric suppression of energy may be larger, and as such reduce energy intake more efficiently.

Minor comments:

  1. Line 339: ‘investigated’ should be ‘investigate’.
  2. Table 3, row 3: should ‘7%’ for energy intake reduction be ‘17%’, in line with the statement in line 253?
  3. The authors may want to check their tables for consistency, in terms of use of capital letters, use of bold text, the use of ‘+’ when there is an increase in energy intake (table 5, combination infusions).

Author Response

Comments reviewer 1

This review aims to summarize the literature on regional effects of GI stimulation on the negative feedback mechanisms that control gastric emptying and gut motility, and appetite and energy intake. The authors look at effects of region, potential differences of nutrients on EI suppression, and the use of non-caloric tastants to suppress energy intake. The review is concise, addresses clearly stated research questions, and is well written. The review contributes novel data to the literature on the gastrointestinal regulation of appetite and energy intake.

I have a few comments that the authors may want to consider.

  1. One major issue is that, although the authors state that a suppression of energy intake as low as 6-7% is clinically relevant, they do not consider the energetic content of the nutrient infusions itself in their statements. If the energy content of the nutrient infusion is identical to, or larger than the suppression in energy intake at the subsequent meal, the strategy to ‘preload’ people with nutrients aiming to reduce their energy intake is arguably not feasible as a potential weight management strategy. Could the authors please include statements on the difference in suppression between the energy content of the infusion and the suppression of energy intake in the studies they included?

Answer:

We agree with the reviewer that the energy content of the infused nutrients should be taken into account in the net reduction in energy intake. In most of the studies analyzed for this review, data on caloric content of the infusions have been provided but the net reduction has not been calculated in the original studies. Therefore we decided to provide more detailed information in the revised manuscript. Data on 1) reduction in energy intake of meal compared to control condition 2) energy content of the nutrients infused and c) net reduction of caloric intake are provided per study/reference in three additional tables: one for fat/fatty acids, one for protein/amino acids and one for carbohydrates (Additional Tables 1-3). When interpreting these results and data, it is important to realize that we are not informed if the nutrients infused for brake activation were completely or only partially digested and absorbed. In fact, in more distal delivery, digestion and absorption may not be complete so that when calculating the ‘net reduction’, the net reduction in energy in fact is underestimated in these conditions. These items are presented and discussed in more detail in the manuscript on 6.1. page 5 and on page 10, line 386-8.

In Table 4 we report, at an aggregated level, on the reduction in energy intake in response to infusion by equicaloric amounts of macronutrients compared to control conditions. Note that the reduction in energy intake is presented as percentage reduction of energy intake as compared to control conditions and does not represent the net reduction in caloric intake, as caloric load of the infused nutrients has not been taken into account in this calculation.

We added to the revised manuscript an additional Table 5, where the net reduction in caloric intake (reduction in energy intake of the meal minus caloric content of the infused nutrients) is presented summarizing in a concise overview the data of the individual studies given in the Additional Tables 1-3.

  1. The authors compare studies with very different protocols to describe their findings. Would the authors consider to include a statement on the nature of the studies included? For example, did the papers include similar participant groups in terms of health, gender, age – and if not, were there differences in findings between the outcomes of the studies? For example, older and obese participants may be less sensitive to the suppressive effect of nutrient infusion on EI, and therefore, may show different results from studies performed in younger/healthy participants. Furthermore, how homogenous are the studies in terms of interventions used? For example, did the described studies use the same sources of proteins/fat for infusion or not (the authors mention this for carbohydrates but not protein, AA, and fat)?

Answer:

The reviewer is correct in asking for more details on the nature of the included studies. We have added to the revised manuscript a Methods section and in this section we included the following statement:   

The included studies all relate to acute or short term (single day to several days experiments) intervention studies in humans with either duodenal or jejunal or ileal intubation with intestinal perfusion of nutrients. In the reported publications young, healthy volunteers, usually of male gender and aged 20-30 years with BMI in the normal range (20-25 kg/m2) have been studied. In the current review we specifically report if studied participants were different (obese versus non-obese and younger versus older participants). Different types of stimuli have been used: long chain fats and fatty acids, protein or amino acids and carbohydrates. The type of stimuli that have been used are listed in Tables 1-3 and for each study separately in the Additional Tables 1-3.”

  1. Furthermore, there are some studies that directly compare the effect of region of infusion on energy intake (ref 10, 41, 47, 48 – maybe others?) within their study. Do these studies show differences between regions of infusion on suppression of EI? Would the authors consider to briefly mention these studies in the text, as these studies would have used the same study protocols for each infusion region, and therefore, may be more appropriate to compare effects between regions than the comparison of studies with different protocols, nutrients loads and/or interventions?

Answer:

The reviewer highlights a relevant issue. Below we have listed the data of studies that investigated the net effect of energy intake reduction with comparison per location within the same study/protocol. This table has not been added to the manuscript but a statement is made on page 9, paragraph 1, line 364-371 that the data of the studies that directly compared the effects of region infusion on energy intake are in line with the general overviews in Tables 1-5.

Table:  Comparison of net effect of energy intake reduction: reduction in energy intake of a meal minus energy content of infusate of fat, proteins or carbohydrates with comparison per location within the same study.

Reference

Location

Infusate:         Energy content          type                 of infusate

Reduction in Energy Intake (EI) of meal

Net effect: reduction EI meal - EI infusate

41

Duodenum

Jejunum

Ileum

casein

casein

casein

60 kcal

60 kcal

60 kcal

+20 kcal

40 kcal

80 kcal

-

-

+

47

Duodenum

Jejunum

glucose

glucose

90 kcal

90 kcal

+160 kcal

-

-

48

Duodenum

Ileum

glucose

glucose

56 kcal

56 kcal

58 kcal

119 kcal

=

+

10

Jejunum

Ileum

corn oil

corn oil

370 kcal

370 kcal

1100 kcal

650 kcal

++

++

33

Duodenum

Ileum

rapeseed oil

rapeseed oil

54 kcal

54 kcal

14 kcal

18 kcal

-

-

19

Duodenum

Ileum

canola oil

canola oil

54 kcal

54 kcal

ileum vs duo:          76 kcal

+

-: reduction EI meal < EI infusate

=: reduction EI meal = EI infusate

+/++/+++: reduction EI meal > EI infusate.  + 0-100 kcal; ++100-300 kcal; +++ : > 300 kcal

Note that + in the section Reduction in Energy Intake (EI) of meal means an increase in energy intake compared to control condition

  1. Figure 1: this figure is not entirely clear to me. Where does the arrow to the colon refer to? And should the arrow from the ileum point to the word ‘duodenum’?

Answer:

We have changed Figure 1 as the arrow is depicting the gastrointestinal tract and have moved the word colon in order to have the figure more clearly depicted.

  1. Lines 237-240: Which amino acids were looked at in the papers included? Are there any differences between the AA’s included?

Answer:

The amino acids studied in the papers included L-tryptophan and leucine. All individual nutrients used for the infusions are listed in Tables 1-3 and in the Additional Tables 1-3. We have changed the manuscript and report on these different amino acids and the results. See page7 line 277-286.

  1. Lines 362-369: the authors talk about the opportunities for applications of their findings. From their findings, are there any nutrients that are more relevant to encapsulate than others? For example, the delivery of carbohydrates, especially in the form of sugar, may not be suitable as it may come with potential negative effects on blood glucose regulation, especially in those who are at risk to develop diabetes. On the other hand, non- or low- caloric compounds may be more feasible as the difference between the caloric load of the compound delivery and caloric suppression of energy may be larger, and as such reduce energy intake more efficiently.

Answer:

We have added the following text, page 12 line 472-477: “When considering whether one specific macronutrient would be most suitable for encapsulation one should keep in mind the effect on energy intake reduction from the different nutrients (Tables 4 and 5). The effect of reduction of energy intake is dependent on the location of release (ileal versus duodenal) rather than on a specific nutrient, as for ileal release all nutrients appear to have similar effects.”

In addition, we added a reference [58], a study from our group that evaluated the effect of an encapsulated carbohydrate-protein mix on energy intake in healthy volunteers. With respect to carbohydrates in the form of sugar: more distal (ileal) delivery of sugars (glucose) will lead to an increase in GLP-1 release with subsequent insulin release and increase in insulin sensitivity, factors that are beneficial in overweight and type 2 diabetes mellitus. In general, for encapsulation the more “energy dense” compounds (fat, fatty acids) are of particular interest. 

Minor comments:

  1. Line 339: ‘investigated’ should be ‘investigate’.

Answer: we have rephrased the sentence according to the reviewers suggestion.

  1. Table 3, row 3: should ‘7%’ for energy intake reduction be ‘17%’, in line with the statement in line 253?

Answer: we changed the table with the correct number according to the reviewers suggestion.

  1. The authors may want to check their tables for consistency, in terms of use of capital letters, use of bold text, the use of ‘+’ when there is an increase in energy intake (table 5, combination infusions).

Answer: we thank the reviewer for this suggestion and have made several changes to the tables for consistency.

Reviewer 2 Report

This is an interesting review of the regional effects GI luminal stimulation. The questions for the review are clearly stated and are appropriate to to the topic. There is a great deal of information and nice use of tables to illustrate the findings. 

It is not clear that this is a systematic review. It would be helpful for the authors to use the PRISMA checklist. The title does not reflect the report as a systematic review. There is limited rationale for the need for the review in this area. 

The methods of the review are missing including inclusion and exclusion criteria, databases, registries, and other reference lists used, and filters or limits used.  The method used to decide which studies are included or excludes should be specified. The methods by which the data were collected is needed. 

A description of the synthesis methods is missing. While there is good information provided, the assessment of risk of bias in the studies is missing.  The tables are helpful but it would be more helpful to note the effect measures. The other criteria from PRISMA related to synthesis methods should be addressed.

There are a number of statements that are not referenced so it is not clear where the information is coming from.  Examples of these are on page 3, lines 101-122 but there are others as well.

The content was not evaluated closely as it is not clear how materials were chosen for the review so the completeness and bias cannot be evaluated.  

Author Response

Reviewer 2

This is an interesting review of the regional effects GI luminal stimulation. The questions for the review are clearly stated and are appropriate to to the topic. There is a great deal of information and nice use of tables to illustrate the findings. 

It is not clear that this is a systematic review. It would be helpful for the authors to use the PRISMA checklist. The title does not reflect the report as a systematic review. There is limited rationale for the need for the review in this area. 

A description of the synthesis methods is missing. While there is good information provided, the assessment of risk of bias in the studies is missing.  The tables are helpful but it would be more helpful to note the effect measures. The other criteria from PRISMA related to synthesis methods should be addressed. 

Answer: We thank the reviewer for addressing these items. We have been invited to contribute to a special edition of Nutrients entitled: ”Satiety and appetite control - gut mechanisms” with the Chapter “Regional effects of gastrointestinal luminal stimulation on appetite and energy intake: (pre)clinical observations”. We have been publishing in this specific domain for several years and have contributed with a significant number of original human intervention studies. For that reason the editors of this issue, Prof Horowitz and prof Feinle-Bisset have invited us. The reviewer is correct that our review is not a systematic review that is fully compliant with the PRISMA checklist. We apologize for not having included any information on methodology, search terms and inclusion and exclusion criteria. This information has now been added to the revised manuscript.

The methods of the review are missing including inclusion and exclusion criteria, databases, registries, and other reference lists used, and filters or limits used.  The method used to decide which studies are included or excludes should be specified. The methods by which the data were collected is needed. 

Answer: see Methods section in revised paper. We performed a search in PubMed for all published articles on intestinal brake mechanism and the effect on energy intake. We used several terms for brake and location of the brake mechanism and effect on energy or food intake in order to be complete and identify all relevant studies for the current review. We used as limits human studies and English language. We have clarified in the manuscript the methods used and describe how we derived the articles.

There are a number of statements that are not referenced so it is not clear where the information is coming from.  Examples of these are on page 3, lines 101-122 but there are others as well.

Answer: We have made several changes to the manuscript and have put more references in the text revised manuscript.

The content was not evaluated closely as it is not clear how materials were chosen for the review so the completeness and bias cannot be evaluated.  

Answer: For the item raised by the reviewer, we refer to our previous answer on the methods.

Round 2

Reviewer 1 Report

Thank you to the authors for their revisions of their review paper. I think the quality and completeness of the paper has improved significantly – well done. Some minor points before the manuscript is acceptable for publication: 1. I suggest that the authors include the table with direct comparisons of the effect of region of infusion in the manuscript, especially since the authors state in the manuscript that ‘this represents the most valid comparison’. I agree that this comparison may be more relevant than comparisons between studies with different study designs and as such the inclusion of this table (either in manuscript or supplementary file) is recommended. 2. When the authors mention ‘energy intake reduction’ throughout the paper, I assume they mean the reduction compared to the non-caloric control condition, but this isn’t always clear, for example in line 369-370 in the titles/legends of tables 4-5 and title/legends/headings of additional tables. Please include this information in the manuscript for clarity. 3. Line 278-282: information in terms of participants/EI suppression is given for studies that infused AA’s but this information is not included for the other nutrients – either include for all sections or delete here for consistency. 4. Lines 488-491: a reference is missing

Author Response

Reply to reviewers comments on manuscript “Nutrients-1162085 R1” entitled “Review on the regional effects of gastrointestinal luminal stimulation on appetite and energy intake: (pre)clinical observations”.

Reviewer 1

Thank you to the authors for their revisions of their review paper. I think the quality and completeness of the paper has improved significantly – well done. Some minor points before the manuscript is acceptable for publication:

  1. I suggest that the authors include the table with direct comparisons of the effect of region of infusion in the manuscript, especially since the authors state in the manuscript that ‘this represents the most valid comparison’. I agree that this comparison may be more relevant than comparisons between studies with different study designs and as such the inclusion of this table (either in manuscript or supplementary file) is recommended.

Answer:

We have added the table with comparison of net effect on energy intake reduction as Table 5 in the manuscript.

  1. When the authors mention ‘energy intake reduction’ throughout the paper, I assume they mean the reduction compared to the non-caloric control condition, but this isn’t always clear, for example in line 369-370 in the titles/legends of tables 4-5 and title/legends/headings of additional tables. Please include this information in the manuscript for clarity.

Answer:

We have added in the manuscript a statement to be more clear on this topic. Additionally we have added legends in the tables to clarify this issue: “Energy intake reduction is presented as percentage reduction of energy intake as compared to control conditions”

  1. Line 278-282: information in terms of participants/EI suppression is given for studies that infused AA’s but this information is not included for the other nutrients – either include for all sections or delete here for consistency.

Answer:

We have removed the information on participants in the section as highlighted by the reviewer, as we state in the methods section that all studies were performed on young healthy male volunteers, and specify where this is different. 

  1. Lines 488-491: a reference is missing

Answer:

We have added the reference.

Reviewer 2 Report

Authors have adequately addressed all concerns. 

Author Response

Reply to reviewers comments on manuscript “Nutrients-1162085” entitled “Review on the regional effects of gastrointestinal luminal stimulation on appetite and energy intake: (pre)clinical observations”.

Reviewer 2

Authors have adequately addressed all concerns. 

Answer: We thank the reviewer for the comment.

Round 3

Reviewer 1 Report

I have no further comments and accept the manuscript for publication in its current form. Congratulations to the authors on an informative paper.